# Clonal Spread of Hospital-Acquired NDM-1-Producing *Klebsiella pneumoniae* and *Escherichia coli* in an Italian Neonatal Surgery Unit: A Retrospective Study

**DOI:** 10.3390/antibiotics12040642

**Published:** 2023-03-24

**Authors:** Marilena Agosta, Daniela Bencardino, Marta Argentieri, Laura Pansani, Annamaria Sisto, Marta Luisa Ciofi Degli Atti, Carmen D’Amore, Pietro Bagolan, Barbara Daniela Iacobelli, Mauro Magnani, Massimiliano Raponi, Carlo Federico Perno, Francesca Andreoni, Paola Bernaschi

**Affiliations:** 1Microbiology and Diagnostic Immunology Unit, Department of Diagnostic and Laboratory Medicine, Bambino Gesù Children’s Hospital, IRCCS, 00163 Rome, Italy; 2Department of Biomolecular Sciences, University of Urbino “Carlo Bo”, 61032 Fano, Italy; 3Clinical Pathways and Epidemiology Unit, Bambino Gesù Children’s Hospital, IRCCS, 00165 Rome, Italy; 4Neonatal Surgery Unit, Medical and Surgical Department of the Fetus-Newborn-Infant, Bambino Gesù Children’s Hospital, IRCCS, 00165 Rome, Italy; 5Health Directorate, Bambino Gesù Children’s Hospital, IRCCS, 00165 Rome, Italy; 6Clinical Pathology Unit, Azienda Sanitaria Territoriale, 61029 Urbino, Italy

**Keywords:** Enterobacterales, carbapenem-resistance, neonates, hospital-acquired colonization, plasmid-typing, sequence type

## Abstract

This article reports a rapid and unexpected spread of colonization cases of NDM-1 carbapenemase-producing *Klebsiella pneumoniae* and *Escherichia coli* in a neonatal surgical unit (NSU) at Bambino Gesù Children’s Hospital in Rome, Italy. Between the 16th of November 2020 and the 18th of January 2021, a total of 20 NDM-1 carbapenemase-producing *K. pneumoniae* (*n* = 8) and *E. coli* (*n* = 12) were isolated from 17 out of 230 stool samples collected from neonates admitted in the aforementioned ward and time period by an active surveillance culture program routinely in place to monitor the prevalence of colonization/infection with multidrug-resistant Gram-negative microorganisms. All strains were characterized by antimicrobial susceptibility testing, detection of resistance determinants, PCR-based replicon typing (PBRT) and multilocus-sequence typing (MLST). All isolates were highly resistant to most of the tested antibiotics, and molecular characterization revealed that all of them harbored the *bla*_NDM-1_ gene. Overall, IncA/C was the most common Inc group (*n* = 20/20), followed by IncFIA (*n* = 17/20), IncFIIK (*n* = 14/20) and IncFII (*n* = 11/20). MLST analysis was performed on all 20 carbapenemase-producing Enterobacterales (CPE) strains, revealing three different Sequence Types (STs) among *E. coli* isolates, with the prevalence of ST131 (*n* = 10/12; 83%). Additionally, among the 8 *K. pneumoniae* strains we found 2 STs with the prevalence of ST37 (*n* = 7/8; 87.5%). Although patient results were positive for CPE colonization during their hospital stay, infection control interventions prevented their dissemination in the ward and no cases of infection were recorded in the same time period.

## 1. Introduction

Worldwide, Enterobacterales are reported as highly resistant to carbapenems, which are last-resort antibiotics largely used to treat severe Gram-negative infections resistant to other antibiotics [1]. One of the main mechanisms used by Enterobacterales to escape the action of carbapenems is the production of carbapenemases (β-lactamases that hydrolyze carbapenems) [1,2].

According to Amber classification, based on amino acid sequences, carbapenemase enzymes are grouped into three classes (A, B and D). Class A includes enzymes that hydrolyze all β-lactams, including monobactams and carbapenems, and they are found in Enterobacterales, *Pseudomonas aeruginosa* and *Acinetobacter* spp. Class B carbapenemases are Metallo-β-lactamases (MBL) and provide resistance to all β-lactams, and they include, among others, VIM (Verona integron-encoded metallo-β-lactamase) and NDM (New Delhi metallo-β-lactamase). Finally, class D is composed of several variants of oxacillin-hydrolyzing β-lactamases (OXAs) that hydrolyze penicillin and meropenem, but not extended-spectrum cephalosporins and aztreonam [3,4].

Many of the genes encoding for carbapenemase are located on plasmids, which facilitates the transfer among bacteria; hence, the spread of this type of resistance [5,6]. In light of this high transmissibility, their early detection achieves more and more importance in limiting the distribution in healthcare settings. Surveillance programs are particularly required for carbapenemases-producing *Klebsiella pneumoniae* (KPC-Kp) and *Escherichia coli* (CP-Ec), frequently responsible for outbreaks as described by the last report of the European Antibiotic Surveillance Network (EARS-Net). KPC-Kp still plays an important role in carbapenem-resistance diffusion, but recent outbreaks were caused by strains carrying *bla*_NDM-1_ and *bla*_OXA-48_ genes, highlighting the concomitant increase in transmissibility and antimicrobial resistance that, in turn, increases the risk for the patient population [7]. Particular concern was for the New Delhi metallo-β-lactamase (NDM), firstly isolated in 2008 from a Swedish patient who returned from travel in India infected with *K. pneumoniae* and *E. coli*. Since then, Enterobacterales harboring *bla*_NDM-1_ have been reported worldwide [8,9]. Notwithstanding that the incidence remains lower than KPC-Kp infections, there was a small but significant rise in invasive infections caused by CP-Ec. The potential risk associated with this issue is that the transmission of these strains in the community where *E. coli* is widely circulating may contribute to the failure of carbapenems treatment for infections [7].

To date, the rapid emergence and worldwide dissemination of carbapenemase-producing Enterobacterales (CPE) is a matter of concern for public health, representing a critical challenge for clinicians [2]. Indeed, the global rise in CPE results in increased healthcare costs, prolonged hospitalization, failure of infection treatment and mortality rate [10,11]. This is particularly serious for neonatal intensive care units (NICU) where the colonization of pediatric patients is recurrent [4,12,13]. Indeed, the frequent occurrence of common infections in children increases their exposure to selective antimicrobial pressure, and, consequently, failure of treatment [14]. Decreased sensitivity to carbapenems might be due to the production of carbapenemases encoded by genes located on mobile elements. Furthermore, the quick molecular evolution of CPE in terms of plasmid profiles contributes to the rapid dissemination of carbapenem-resistance genes among strains [5,6]. Of particular concern are the community-acquired (CA) pathogens introduced in the hospital through hospitalized neonates and infants themselves, their parents and also healthcare professionals. In this context, the movements of patients from and to different NICUs plays an important role, due, for instance, to previous exposure to high-risk care settings or antibiotic therapies facilitating the spread of resistant strains [14,15]. Additionally, prolonged hospital stays due to infections that are difficult to treat is an added well-known risk factor for hospital-acquired colonization (HAC), probably due to the patients sharing the same spaces (i.e., room, bathroom, medical examination rooms that could be potential colonization sites) [16,17]. In recent years, resistance to carbapenems in the pediatric population has dramatically increased; therefore, addressing this threat is urgent.

Hospitals need good infection control and, in this sense, several guidelines have been adopted to prevent and monitor the spread of CPE [7,18]. Active surveillance based on culture and molecular typing, adoption of contact precautions, isolation or cohorting, and when necessary, decolonization, are the current prevention strategies implemented to monitor carriage and infection. However, the surveillance of the healthy population is also strongly suggested to identify asymptomatic carriers responsible for the spread of multidrug-resistant organisms (MDROs), especially among individuals considered at risk, such as pediatric ones [19,20]. In this study, we reported a sudden clonal diffusion of colonization cases by NDM-1-producing *K. pneumoniae* and *E. coli* in an Italian NICU between November 2020 and January 2021. We characterized strains in order to study their molecular features in terms of antibiotic resistance, plasmid profiles and circulating clones.

## 2. Results

### 2.1. Epidemiological and Clinical Investigation

Between November 2020 and January 2021, 230 stool samples were collected from newborn patients admitted to NSU for routine analysis dedicated to detect MDROs intestinal colonization status.

Of these, 17 of the stool cultures, collected from 9 females and 8 males aged from 12 days to 1 year (median (IQR) age was 30 (16–75) days), tested positive for CPE colonization and 20 CPEs were isolated belonging only to *K. pneumoniae* and *E. coli species*. Information concerning clinical characteristics of patients available for this study is collected in Table 1.

Of the strains, 90% (*n*/N = 18/20) were isolated from 16 Italian patients from the geographical area of Rome, whereas the remaining 10% (*n*/N = 2/20) were both isolated from 1 patient coming from Romania. From all patients, 1 strain was isolated from each stool culture, with the exception of 3 newborns in which 2 different CPEs were isolated from the same stool sample. The stool cultures of all patients were negative for MDROs colonization at the time of their admission but positive after 48 h (starting from a minimum time of 6 days up to 212 days from admission, i.e., the time of entry to the hospital). Hence, all patients acquired CPE colonization during their hospital stay.

The first case of a positive surveillance stool culture was in a neonate who had been admitted to the NSU on the 8th of October for the presence of a left lateral cervical neoformation from birth. He tested positive for *E. coli*-NDM-1 colonization on the 16th of November. Infection prevention and control measures were implemented promptly, and the neonate remained under contact isolation precautions until discharge. Negative evaluation of the stool cultures monitored took place on the 21st of December. However, on the 14th of December, 3 more neonates (1 had been in NSU since June because of prematurity and intestinal perforation, always colonization negative until that time) were found to be colonized with NDM-1 carbapenemase-producing *K. pneumoniae* (*n* = 2) and *E. coli* (*n* = 1), respectively. Thereafter, between the 21st of December 2020 and the 18th of January 2021, 13 further neonates acquired NDM-1 *K. pneumoniae* (*n* = 6) and *E. coli* (*n* = 10) colonization, for a total of 16 isolates (Figure 1).

When surveillance stool culture results were positive for CPE colonization, patients were immediately placed in contact isolation; however, despite the adoption of preventive measures, in the following weeks, out of the stool cultures returned, only 3 cases were negative. In all other cases, the surveillance stool cultures remained positive for NDM-1 carbapenemase-producing *K. pneumoniae* or *E. coli*.

It should be noted that no infection occurred and no patient experienced complications of their underlying pathological conditions. For this reason, patients were gradually discharged from the ward according to their clinical improvement. The last neonate was discharged from NSU on the 20th of March 2021. He was colonized by NDM-1 carbapenemase-producing *K. pneumoniae* since the 28th of December 2020, and his surveillance stool culture tested negative for CPE colonization on the 1st of March 2021.

There were no further cases of colonization or infection by NDM-1 carbapenemase-producing Enterobacterales in the NSU during the following months. After a short period (February–March) of absence of further isolations, a sporadic circulation resumed in April 2021, with the isolation of 13 new cases of NDM-1 carbapenemase-producing Enterobacterales during the whole of 2021. No cases were registered in 2022 and, to date, no more NDM-producing Enterobacterales have been isolated in the ward.

### 2.2. Antimicrobial Susceptibility Patterns and Resistance Determinants

As reported in Table 2, all isolates were highly resistant to most of the tested antibiotics, including carbapenems. In particular, all were resistant to ertapenem and 78% were resistant to imipenem and meropenem. Additionally, all *E. coli* and *K. pneumoniae* revealed high resistance towards amikacin, amoxicillin-clavulanic acid, aztreonam, cefotaxime, ceftazidime, ceftazidime/avibactam, ceftolozane/tazobactam, gentamycin, ertapenem, piperacillin/tazobactam and tobramycin. All strains were susceptible to colistin and tigecycline whereas almost half of them were susceptible to ciprofloxacin and trimethoprim-sulfamethoxazole. All isolates tested positive for NDM-type carbapenemase by immunochromatographic assay, and the presence of the encoding *bla*_NDM_ gene was confirmed by PCR-based method. 

Finally, amplicon sequencing revealed that all of them harbored the *bla*_NDM-1_ gene. Despite this, the phenotypic analysis showed different resistance patterns among the strains. In particular, a high variability was observed among the 12 strains of *E. coli* with the detection of 11 different resistance patterns; on the contrary, in *K. pneumoniae* collection, this variability was more limited (only three different patterns were observed). All these data highlight the serious implications that these circulating strains could have in neonatal wards.

### 2.3. Plasmid Profiles and Multilocus Sequence Typing

A summary of replicons detected among the 20 CPE strains is given in Figure 2. Overall, IncA/C was the most common Inc group (*n* = 20/20), followed by IncFIA (*n* = 17/20), IncFIIK (*n* = 14/20) and IncFII (*n* = 11/20). In this study, 7 out of 30 replicons were identified (IncX1, IncFIB, IncA/C, IncFII, IncFIA, IncFIIK, IncM). Half of the isolates (*n* = 10/20) were characterized by the multi-replicon status carrying 3 or more different Inc groups. The predominant multi-replicon profiles were FIA, A/C, FIIK, FII (*n* = 7) and FIA, A/C, FIIK (*n* = 6), both found only among *E. coli* strains and *K. pneumoniae*, respectively.

MLST analysis was performed on all 20 CPE strains, which revealed 3 different STs among *E. coli* isolates, with the prevalence of ST131 (*n* = 10; 83%), followed by 1 strain belonging to ST155 and another to ST101. Additionally, among the 8 *K. pneumoniae* strains, we found 2 STs with the prevalence of ST37 (*n* = 7/8), followed by 1 strain belonging to ST160. All were isolated from patients colonized during their hospitalization in the neonatal surgery unit. Moreover, in both *K. pneumoniae* and in *E. coli* strains, 2 replicon profiles were associated with the same prevalent ST (ST37 and ST131 in *K. pneumoniae* and in *E. coli*, respectively). These findings suggest the circulation of the same clones among patients hospitalized in the same ward.

## 3. Discussion

Here, we described the detection and containment of NDM-1-producing Enterobacterales colonization within the neonatal ward of a pediatric hospital in Central Italy. In both *K. pneumoniae* and *E. coli* collection we found isolates genetically related because they belonged to the same clonal groups, within which are strains with the same antimicrobial resistance patterns and plasmid replicon types. Since November 2018, the rapid and increased diffusion of NDM-producing Enterobacterales isolates was recorded in Italy, starting from the Tuscany Region, leading to the establishment of a multidisciplinary regional task force to manage the health emergency [21].

During recent decades, several screening programs were implemented in clinical settings worldwide. Many cases of colonization were described, which highlighted the risk for fragile or immunocompromised patients [22,23,24]. Few studies have been carried out to evaluate the dissemination of CPE among neonatal patients in Italy, but in all of these cases KPC-producing strains emerged [25,26,27,28]. To the best of our knowledge, the present study and another previously published by our team [12], represent the only investigations describing the circulation of NDM-producing *K. pneumoniae* and *E. coli* in Italian NICUs between July 2016 and December 2019. Since January 2020, the circulation of NDM-producing Enterobacterales gradually decreased, with a total of 22 new NDM-colonization cases in NSU (prevalence of 78% among 28 CPE isolated before the 16th of November). In the present study, the detection of three patients positive for NDM-intestinal colonization on the 14th of December 2020 was the starting point of the clonal spread. Since then, a retrospective analysis of the medical records of the patients admitted in the ward allowed to identify the index case in a patient colonized with NDM-producing *E. coli* a month earlier, on the 16th of November. For this patient, isolation and adoption of adequate contact precautions turned out to be effective, leading to the negativization of the surveillance stool culture on the 21st of December. However, it should be noted that this neonate remained persistently positive for intestinal colonization for over a month. This probably made it difficult to maintain adequate contact precautions over such a long period of time and it may have contributed to the spread of colonization to other patients in the ward. In fact, intervention by designated staff for each neonate was difficult to apply due to the extreme fragility and complex condition of these patients that could often be managed by a team of highly qualified and experienced personnel and not by a single healthcare worker.

Beyond the use of dedicated patient equipment, healthcare workers and newborns’ parents were further educated to standard infection control procedures, including adoption of contact precautions, use of personal protection equipment (i.e., disposable gloves or gowns) and reinforced sanitation of the environment. Hand hygiene was implemented upon entry into the NSU with the use of alcohol-based hand sanitizers, especially after patient contacts. However, alternative sources of contamination and transmission by NDM producing Enterobacterales could not be investigated, including environmental sampling or the possible carriage by healthcare workers or parents. Moreover, risk factors in NSU patients, e.g., low birth weight, prematurity, length of stay, use of invasive devices and antibiotic consumption before admission and during hospitalization to the NSU, were not available in our study. Despite these limitations, and even though the source was not found, this clonal diffusion was interrupted approximately three months later with the negativization of the surveillance stool culture of the last persistently positive patient on the 1st of March 2021. Although patients were positive for CPE colonization during their hospital stay, infection control interventions prevented their dissemination in the ward: no cases of infection occurred in the same time period and no patient experienced complications regarding their underlying pathological conditions.

Despite the increased occurrence of colonization status worldwide, the available literature concerning CPE colonization in NICUs patients is limited and risk factors need to be extensively investigated [4,29]. On the other hand, it is well-known that infectious diseases caused by NDM-producing isolates are strongly associated with a high rate of morbidity and mortality among fragile patients, including newborns, due to limited therapeutic options [30]. In particular, NDM-1-producing *K. pneumoniae* and *E. coli* are responsible for nosocomial infections, such as urinary tract infections, peritonitis, septicemia and pulmonary infections [30,31]. All patients considered in this investigation were hospitalized in the neonatal surgery unit, highlighting their fragile health status for which antibiotic treatment acquires particular significance. In our study, we confirmed the high rate of resistance towards the main antibiotics and that the effective therapeutic options are limited to colistin and tigecycline, considered last-resort antibiotics for the treatment of these infections. However, recent studies described the potential emergence of colistin-resistance in response to both tigecycline and colistin pressure [32], therefore, limiting their use in clinical treatment. Further investigations will be necessary to ascertain this important point, particularly in view of the extensive use of tigecycline in many hospital wards.

It should be noted that *E. coli* ST131 (*n* = 10/12) and *K. pneumoniae* ST37 (*n* = 7/8) were the predominant epidemic clones found in this study. The ST131 in *E. coli* has been frequently reported in clinical environments and largely found to be associated with the carriage of the *bla*_NDM-1_ gene harbored by the IncF group as observed in this study [33]. Although it was prevalent in our collection of NDM-1-producing *K. pneumoniae* isolates, ST37 is not widely described as being associated with this gene. Indeed, the most commonly known NDM-1-positive clones in *K. pneumoniae* are ST14, ST11 or ST147 [34,35,36], whereas ST37 is known to carry the IncF plasmid family, as in our case, but harboring *bla*_OXA_ genes [37,38,39]. However, a limitation of this study is to conclude the clonal spread on the basis of MLST. Indeed, the novel applications of whole-genome sequencing (WGS) are more discriminating and informative on relatedness and molecular mechanisms than previous molecular typing methods. In addition, WGS would have been useful to collect information on the transmission of genes through plasmids. Taken together, these data suggest the rapid and continuous evolution of circulating clones, indicating that, in the absence of effective therapy, surveillance acquires increased value to prevent and control the diffusion of NDM-producing isolates with increased pathogenic profiles in neonatal wards. There is reason to believe that the continuous surveillance of CPE-colonized neonates adopted in our hospital prevented further cross-transmission and progression of colonization and infection in more patients.

Finally, we underline that the vulnerability to colonization or infection with CPE among newborns makes it increasingly necessary to adopt control measures, including, for instance, hand hygiene, contact precautions and cohort nursing care. All of these could be very useful to reduce cross-infection and avoid the rapid spread or clonal dissemination of serious clones circulating in healthcare facilities.

## 4. Materials and Methods

### 4.1. Ethics Statements

Only bacterial isolates recovered from routine screening and diagnostic laboratory tests were assessed in this study without direct use of clinical specimens. Considering the retrospective nature of the analysis, the current study did not require the approval of the local ethics committee according to current legislation, but a notification was sent. In addition, parents or legal guardians of patients provided consent to use personal data for diagnosis, treatment and related research purposes. Data were retrospectively analyzed in line with personal data protection policies and patient consent was not required.

### 4.2. Study Design

This study was a single-center, retrospective investigation carried out on newborn patients admitted to the neonatal surgery unit (NSU) of Bambino Gesù Children’s Hospital in Rome (Italy) between November 2020 and January 2021. The collection of samples was followed by a long period of monitoring to detect the potential reoccurrence of colonization cases as described by the surveillance program of the Hospital. We evaluated all consecutive newborn patients admitted to the NSU, and a total of 230 stool samples were collected to detect MDROs intestinal carriers, according to the active surveillance protocol issued by the hospital infection control committee. 

We collected demographic and clinical information from electronic medical records about patients, the results of which were positive for CPE intestinal colonization, including sex, age, geographical origin, the date of stool sample collection and the preventive measures adopted to avoid the occurrence of infection. After isolation and identification, all CPE strains were characterized by antimicrobial susceptibility testing, detection of carbapenemase resistance determinants and PCR-based replicon typing (PBRT) at the microbiology laboratory of Bambino Gesù Children’s Hospital. Finally, to evaluate their clonality, all strains of *K. pneumoniae* and *E. coli* were analyzed by multilocus-sequence typing (MLST) at the molecular biology laboratory of Urbino University.

### 4.3. Hospital Setting and Preventive Measures

Bambino Gesù Children’s Hospital, with its 607 beds and 26,179 ordinary annual admissions, is the largest university hospital and pediatric research center in Europe, acting as the point of reference for the health of children and teenagers coming from all over Italy and around the world. The NSU assists infants and newborns with major congenital anomalies that can be corrected with surgery, mainly thoracic and abdominal diseases. NSU had 19 inpatient beds; the average annual number of inpatient admissions and surgical procedures with advanced technologies were equal to approximately 350 and 500, respectively. Since 2012, a routine surveillance protocol has been in place to monitor the prevalence of colonization/infection with multidrug-resistant Gram-negative microorganisms. Gram-negative microorganisms included in the surveillance screening protocol were carbapenemase-producing Enterobacterales (CPE) and carbapenemase-producing or Extensively Drug Resistant (XDR) *Pseudomonas aeruginosa* and *Acinetobacter baumannii*

Colonization status was defined as the presence of multidrug-resistant organisms (MDROs) in the gastrointestinal tract or in other non-sterile sites such as the upper respiratory tract, but patients did not demonstrate signs and/or symptoms of infection and antibiotic treatment was not recommended. According to the active surveillance protocol issued by the hospital infection control committee, screening of MDROs carriers is performed routinely from stool samples collected at the time of admission and then weekly, until patients are discharged or in the case of the positivity of two consecutive samples. In the case of patients testing positive for MDROs colonization for the first time, active surveillance of patients admitted to the same ward is initiated, with weekly stool culture sampling. Weekly screening of all patients is conducted until no further transmission is detected in the affected unit and patients with confirmed MDROs colonization/infection have been placed in isolation under contact precautions, within the past three weeks. On the 16th of November 2020, the first CPE was identified from a surveillance stool culture of a newborn patient admitted to NSU. This prompted the initiation of active surveillance of all patients in the ward, according to the above described procedures, which led to the identification of 19 additional cases of hospital-acquired colonization.

### 4.4. Bacterial Isolation, Identification and Antimicrobial Susceptibility Testing

For cultural screening, each swab was inoculated on a set of two plates: a CHROMID^®^ CARBA plate and a MacConkey agar plate (bioMérieux, Lyon, France) with a 10 μg meropenem disk (Oxoid, UK). Plates were incubated at 37 °C overnight. All the morphologically different colonies growing into the meropenem disk halo (zone diameter <28 mm) and on the selective chromogenic medium were picked up and sub-cultured for purity onto a MacConkey agar plate (bioMérieux, Lyon, France). Isolated colonies were identified by using Matrix-Assisted Laser Desorption Ionization–Time-of-Flight Mass Spectrometry (MALDI-TOF, Bruker Daltonics, Bremen, Germany) and tested for their antimicrobial susceptibility by broth microdilution method, using the Sensititre Gram Negative DMKGN plate (ThermoFisher Scientific, CA, USA). According to the manufacturer’s instructions, isolated colonies were diluted into Sensititre Sterile Water to measure a 0.5 McFarland standard, and 30 µL of this suspension was inoculated into a Sensititre Muller Hinton Broth tube. Then, each well of the plate was inoculated with a 50 µL volume of the broth suspension using a multi-channel pipette and the prepared plate was incubated overnight between 34 and 36 °C in a non-CO_2_ incubator.

The following antimicrobial agents were tested, and their growth endpoint were read manually: amikacin (AMK), amoxicillin-clavulanic acid (AMC), aztreonam (ATM), cefotaxime (CTX), ceftazidime (CAZ), ciprofloxacin (CIP), gentamicin (GEN), imipenem (IPM), meropenem (MEM), colistin (CST), piperacillin-tazobactam (TZP), tigecycline (TGC), trimethoprim-sulfamethoxazole (SXT), ertapenem (ETP), tobramycin (TM) ceftolozane/tazobactam (CT), ceftazidime/avibactam (CZA). The isolates were identified as resistant to carbapenems according to clinical breakpoints based on the European Committee on Antimicrobial Susceptibility Testing (EUCAST) breakpoints tables. We adopted updated EUCAST breakpoints tables (version 10.0 and version 11.0, for 2020 and 2021, respectively) (https://www.eucast.org/clinical_breakpoints/, accessed on 24 February 2023).

### 4.5. Immunochromatographic Assay and PCR-Based Methods for Carbapenemase Genes

The NG-Test Carba 5 immunochromatographic assay (NG Biotech, Rennes, France) was used to detect the resistance determinants. This immunoassay has been developed to rapidly detect the five main carbapenemases, i.e., KPC, OXA-48-like, NDM, VIM and IMP. Briefly, one single subcultured isolated colony, harvested from a MacConkey agar plate (bioMérieux, Lyon, France) after incubation at 37 °C overnight, was collected from the plate with an inoculation loop and suspended in 150 µL of extraction buffer to perform the lysis step. Then, 100 µL of this extract was loaded on the cassette and allowed to migrate. Time until the appearance of one or more red lines in the test region of the cassette was recorded in comparison to a line in the control region, with the final reading performed at 15 min, according to the manufacturer’s instructions [40,41].

The open reading frame of the *bla*_NDM_ gene was amplified by PCR using two pairs of primer sequences (NDM upstream FW: 5′-CTGCATTTGCGGGGTTTTTA-3′; RV: 5′-CGCCATCCCTGACGATCAAA-3′; NDM-downstream FW: 5′-ATCAAGGACAGCAAGGCCAA-3′; RV: 5′-CTTCCAACTCGTCGCAAAGC-′), following the conditions described by Poirel and colleagues [42]. Successively, all amplicons were sequenced using a BigDye Terminator v. 1.1 Cycle Sequencing kit on an ABI PRISM^®^ 310 Genetic Analyzer (Thermo Fisher Scientific, Waltham, MA, USA). The alignment between sequences and the related reference was carried out using Unipro UGene version 38.0 software [43] and compared using BLAST (basic local alignment search tool).

### 4.6. Bacterial DNA Extraction and Plasmid Typing

The total DNA of isolated colonies, which tested positive for carbapenemase production, was extracted using the EZ1 DNA Tissue Kit (Qiagen, North Rhine-Westphalia, Germany). Briefly, isolated colonies were diluted in 0.45% saline to the turbidity of 0.5 McFarland standard and 200 µL of the suspension were transferred into a 2-mL vial. The vial and prefilled reagent cartridges were loaded onto the EZ1 Advanced XL (Qiagen, North Rhine-Westphalia, Germany) instrument and the protocol for automated purification of bacterial DNA, using magnetic particle technology, was started. DNA was eluted in a final volume of 100 µL and stored at −20 °C until use. Additionally, 1 µL of the extracted DNA was used for plasmid typing using the PCR-based replicon typing (PBRT) kit 2.0 (Diatheva, Fano, Italy). This novel PBRT assay, consisting of 8 multiplex PCRs, is able to detect 30 different replicons of the main plasmid families in Enterobacterales [44]. This kit was used following the manufacturer’s instructions, including positive controls. Amplification products were resolved and visualized directly on a closed ready-to-use 2.2% agarose gel-cassette-system (FlashGel-Lonza, Basel, Switzerland) using the 100 bp FlashGel DNA marker. Fragments obtained were compared to positive controls of each multiplex PCR.

### 4.7. Multilocus Sequence Typing

Multilocus sequence typing (MLST) of *K. pneumoniae* was performed using seven housekeeping genes (*gapA*, *infB*, *mdh*, *pgi*, *phoE*, *rpoB* and *tonB*) in accordance with protocol 2 of the Institute Pasteur Klebsiella MLST database (https://bigsdb.pasteur.fr/klebsiella/primers_used.html, accessed on 24 February 2023). Instead, *E. coli* strains were subtyped considering the seven housekeeping genes (*adk*, *fum*C, *gyr*B, *icd*, *mdh*, *pur*A and *rec*A) selected from the Enterobase MLST database (http://enterobase.warwick.ac.uk/species/index/ecoli, accessed on 24 February 2023). Primer sequences and reaction conditions used to type strains were previously described by Wirth and colleagues (2006) [45].

Amplicons were sequenced using the BigDye Terminator v. 1.1 Cycle Sequencing kit on an ABI PRISM^®^ 310 Genetic Analyzer (Thermo Fisher Scientific, Waltham, MA, USA). All sequences were aligned with related reference through Unipro UGene version 38.0 software [43]. Allele numbers and sequence types (STs) were determined through the corresponding MLST database.

## 5. Conclusions

The circulation of NDM-1-producing *K. pneumoniae* and *E. coli* is a real risk for NICUs with potential complications for neonatal patients. CPE colonization or infection cases in NICUs are a serious problem for newborns worldwide, and surveillance is the crucial aspect for a good infection control program. A systematic approach adopted during the hospital routine is the key to successfully control CPE dissemination and the associated risks.

## Figures and Tables

**Figure 1 antibiotics-12-00642-f001:**
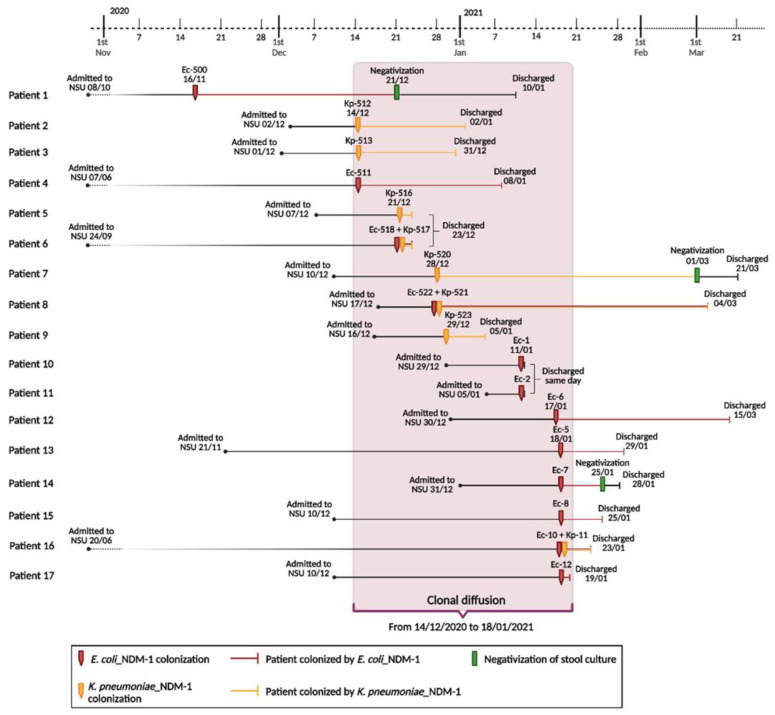
Timeline of clonal diffusion of NDM-1-producing *Escherichia coli* and *Klebsiella pneumoniae* strains (created with BioRender.com, accessed on 26 January 2023. Reproduction of this figure requires permission from Bio.Render.com). Abbreviations: NSU, Neonatal Surgical Unit; Ec, *Escherichia coli*; Kp, *K. pneumoniae*.

**Figure 2 antibiotics-12-00642-f002:**
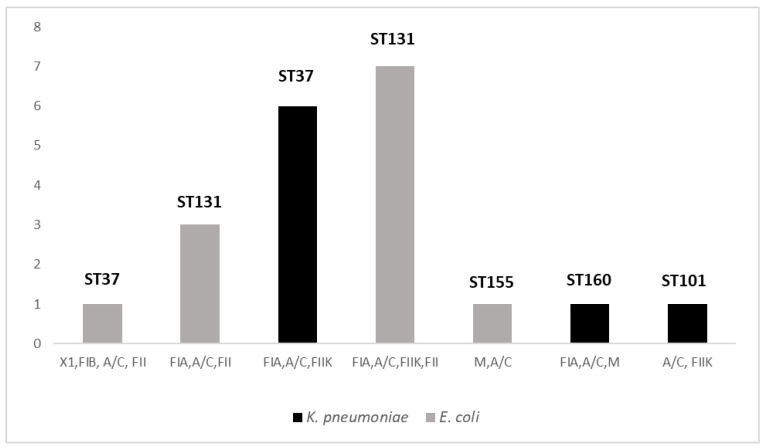
Distribution of replicon profiles and STs among collected strains of *K. pneumoniae* and *E. coli*. Seven replicons profiles were distributed among 20 CPE strains and all were associated with a single species. Five PBRT profiles were composed by three or more Inc groups. Both in *K. pneumoniae* and in *E. coli* strains, two replicon patterns were associated with the same prevalent ST (ST37 and ST131 in *K. pneumoniae* and in *E. coli*, respectively).

**Table 1 antibiotics-12-00642-t001:** Accessible clinical characteristics of patients who tested positive for intestinal colonization by NDM-1 producing *K. pneumoniae* and *E. coli* strains.

Patient N°	Sex ^a^	Age ^b^	Strains ^c^	Date of Admission	Date of Discharge	Isolation of Strains	N° of Days Post-Admission with Positive Stool Culture	Negativization of Surveillance Stool Culture
Patient 1	F	1 mth	Ec-500	8 October 2020	10 January 2021	16 November 2020	39	Yes (21 December 2020)
Patient 2	M	17 days	Kp-512	2 December 2020	2 January 2021	14 December 2020	12	No
Patient 3	F	13 days	Kp-513	1 December 2021	31 December 2021	14 December 2020	13	No
Patient 4	F	7 mths	Ec-511	7 June 2020	8 January 2021	14 December 2020	190	No
Patient 5	F	17 days	Kp-516	7 December 2020	23 December 2021	21 December 2020	14	No
Patient 6	M	2 mths	Kp-517	24 September 2020	23 December 2021	21 December 2020	88	No
			Ec-518					
Patient 7	M	4 mths	Kp-520	10 December 2020	21 March 2021	28 December 2020	18	Yes (1 March 2021)
Patient 8	F	12 days	Kp-521	17 December /2020	4 March 2021	28 December 2020	11	No
			Ec-522					
Patient 9	M	12 days	Kp-523	16 December 2020	5 January /2021	29 December 2020	13	No
Patient 10	F	13 days	Ec-1	29 December 2020	11 January 2021	11 January 2021	13	No
Patient 11	M	1 mth	Ec-2	5 January 2021	11 January 2021	11 January 2021	6	No
Patient 12	M	19 days	Ec-6	30 December 2020	15 March 2021	17 January 2021	18	No
Patient 13	F	5 mths	Ec-5	21 November 2020	29 January 2021	18 January 2021	58	No
Patient 14	F	18 days	Ec-7	31 December 2020	28 January 2021	18 January 2021	18	Yes (25 January 2021)
Patient 15	F	1 mth	Ec-8	10 December 2020	25 January 2021	18 January 2021	39	No
Patient 16	M	1 year	Ec-10	20 June 2020	23 January /2021	18 January 2021	212	No
			Kp-11					
Patient 17	M	1 mth	Ec-12	10 December 2020	19 January 2021	18 January 2021	39	No

^a^ F, female; M, male; ^b^ mth/mths, month/months; ^c^ Kp and Ec are abbreviations for the NDM-1 producing *K. pneumoniae* and *E. coli* strains, respectively.

**Table 2 antibiotics-12-00642-t002:** Genotypic and phenotypic profiles of strains isolated from neonatal patients.

Strains ^a^	Carbapenemase Genes	Pattern of Resistance ^b^	PBRT Profile ^c^	ST
Ec-500	*bla* _NDM-1_	AMK, AMC, ATM, CTX, CAZ, CZA, CT, GEN, IPM, ETP, MEM, TZP, TM	FII, X1, A/C, FIB	101
Ec-511		AMK, AMC, ATM, CTX, CAZ, CZA, CT, CIP, GEN, IPM, ETP, MEM, TZP, TM, SXT	FII, A/C, FIA	131
Ec-6		AMK, AMC, AT, CTX, CAZ, CZA, CT, CIP, GEN, ETP, TZP, TM, SXT		
Ec-522		AMK, AMC, ATM, CTX, CAZ, CZA, CT, CIP, GEN, IPM, ETP, MEM, TZP, TM		
Ec-1		AMK, AMC, ATM, CTX, CAZ, CZA, CT, CIP, GEN, ETP, TZP, TM, SXT	FII, FIIK, A/C, FIA	
Ec-2				
Ec-518		AMK, AMC, CTX, CAZ, CT, CIP, GEN, ETP, TZP, TM		
Ec-5		AMK, AMC, AT, CTX, CAZ, CZA, CT, CIP, GEN, ETP, TZP, TM, SXT		
Ec-7		AMK, AMC, ATM, CTX, CAZ, CZA, CT, CIP, GEN, IPM, ETP, MEM, TZP		
Ec-8		AMK, AMC, ATM, CTX, CAZ, CZA, CT, CIP, GEN, IPM, ETP, MEM, TZP, SXT		
Ec-12		AMK, AMC, ATM, CTX, CAZ, CZA, CT, GEN, IPM, ETP, MEM, TZP, TM		
Ec-10		AMK, AMC, ATM, CTX, CAZ, CT, GEN, IPM, ETP, MEM, TZP	A/C, M	155
Kp-512	*bla* _NDM-1_	AMK, AMC, ATM, CTX, CAZ, CZA, CT, GEN, IPM, ETP, MEM, TZP, TM	FIIK, A/C, FIA	37
Kp-513				
Kp-516				
Kp-517				
Kp-520				
Kp-523				
Kp-521		AMK, AMC, ATM, CTX, CAZ, CZA, CT, CIP, GEN, IPM, ETP, MEM, TZP, TM, SXT	FIIK, A/C	
Kp-11		AMK, AMC, ATM, CTX, CAZ, CZA, CT, GEN, IPM, ETP, MEM, TZP, TM	A/C, FIA, M	160

^a^ Kp and Ec are abbreviations for the *K. pneumoniae* and *E. coli* strains, respectively; ^b^ amikacin (AMK), amoxicillin-clavulanic acid (AMC), aztreonam (ATM), cefotaxime (CTX), ceftazidime (CAZ), ciprofloxacin (CIP), gentamicin (GEN), imipenem (IPM), meropenem (MEM), piperacillin-tazobactam (TZP), trimethoprim-sulfamethoxazole (SXT), ertapenem (ETP), tobramycin (TM), ceftolozane/tazobactam (CT), ceftazidime/avibactam (CZA); ^c^ PBRT (PCR-Based Replicon Typing).

## Data Availability

All data are described within the text.

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
