# Peer review of "Clonal Spread of Hospital-Acquired NDM-1-Producing Klebsiella pneumoniae and Escherichia coli in an Italian Neonatal Surgery Unit: A Retrospective Study"

_antibiotics, 2023, doi:10.3390/antibiotics12040642_

Round 1
Reviewer 1 Report
The authors report a rapid and unexpected spread of colonization cases of NDM-1 carbapenemase-producing Klebsiella pneumoniae and Escherichia coli in a neonatal surgical unit (NSU) in Italy. Overall, the manuscript is technically sound, the data supports the findings, all the data underlying the findings in the manuscript is fully available and the manuscript is presented in an intelligible fashion and written in standard English. The authors might consider revising the manuscript, and the comments are given below.
· Please specify the patients Antibiotic consumption before admission to the NSU, if any.
· Clinical characteristics of patients suggested to present in a tabular form (example given below):
Table format: Patients Number/sex/age/hospital stayed time (admission to release)/date isolate identified/types of infection or main symptoms/Antibiotic therapy/Clinical outcome.
· Please analyse Logistic regression for possession of MDR phenotype among NDM-1-producing Klebsiella pneumoniae and Escherichia coli based on age, sex, origin.
· Did author Whole genome sequence for isolates? If so, suggest Minimum spanning tree for K. pneumoniae and E. coli whole-genome MLST scheme.
· Please describe the infection control measures implemented for patients at NSU in discussion section clearly.
· Suggest to add Gen accession number for the sequence analysed in the text.
· Check the spelling mistakes in the text (for example, check highlights in attached file)
· Note the limitations of the study.
Overall, the manuscript is interesting. Manuscript could be suitable as a SHORT REPORT/CLINICAL REPORT rather than Full length article. It could be publishable after addressing the comments given above.

Author Response
Response to Reviewer 1
- Please specify the patients Antibiotic consumption before admission to the NSU, if any.
We thank the reviewer for this note. As detailed in the “study design” section, only clinical information regarding sex, age, geographical origin, the date of stool sample collection, and the preventive measures adopted to avoid the occurrence of infection were collected from medical records. Any other information, including antibiotic consumption, was not available, as the approval by the Ethics Committee was not requested. We described this limitation of our study as suggested also by another reviewer. See lines 282-285.
- Clinical characteristics of patients suggested to present in a tabular form (example given below):
Table format: Patients Number/sex/age/hospital stayed time (admission to release)/date isolate identified/types of infection or main symptoms/Antibiotic therapy/Clinical outcome.
Thanks for this suggestion. We added Table 1 presenting patients' clinical characteristics accessible for this study (i.e Number/sex/age/hospital stayed time (admission to release)/date isolate identified).
- Please analyse Logistic regression for possession of MDR phenotype among NDM-1-producing Klebsiella pneumoniae and Escherichia coli based on age, sex, origin.
Thanks for this comment. However, we underline that this study was based mainly on the investigation of molecular profiles of strains in order to assess their clonal spread. It was not an epidemiological study for clinical characteristics of patients due also to the limited size of collection. In light of these reasons we did not deem it appropriate to carry out any statistical analyses including logistic regression.
- Did author Whole genome sequence for isolates? If so, suggest Minimum spanning tree for K. pneumoniae and E. coli whole-genome MLST scheme.
We thank the reviewer for this interesting suggestion. However, we did not perform WGS of isolates to determine sequence type (ST). MLST was performed amplifying, and then sequencing, the seven housekeeping genes in accordance with the Institute Pasteur. We did not consider the possibility of building a phylogenetic tree in the light of two reasons: the limited number of strains for both species and most of them share the same alleles. Hence, a minimum spanning tree could be little informative.
- Please describe the infection control measures implemented for patients at NSU in discussion section clearly.
Thanks for this suggestion. We clarified the description of infection control procedures in discussion section as requested. See lines 275-280.
- Suggest to add Gen accession number for the sequence analysed in the text.
Thank you for this observation. Gene accession number of reference sequences were not reported because the Institute Pasteur recommends to use sequences available on its website where different protocols are provided. Furthermore, we obtained known sequence alleles hence no novel submissions in databank were carried out.
- Check the spelling mistakes in the text (for example, check highlights in attached file)
We thank the reviewer for highlighting spelling mistakes that in revised version have been corrected. See lines 265-353-461 of revised version where changes are marked in yellow.
- Note the limitations of the study.
We thank the reviewer for this note. We clarify that a limitation of our investigation is the clonal spread of strains investigated by MLST that is less discriminating and informative than WGS. See lines 316-320 of discussion section.
- Overall, the manuscript is interesting. Manuscript could be suitable as a SHORT REPORT/CLINICAL REPORT rather than Full length article. It could be publishable after addressing the comments given above.
In order to choose the article type we considered instructions provided by Antibiotics and, in light of them, the research paper format was the most appropriate. Indeed, short or clinical reports should include presentation of medical conditions, symptoms, treatment and outcome that do not meet the investigation carried out in our case. However, if the Journal believes that the format of the manuscript should be changed, we will be available for any solution.
Reviewer 2 Report
The original article entitled "Clonal Spread of Hospital-Acquired Colonization Cases by NDM-1-producing Klebsiella pneumoniae and Escherichia coli in an Italian Neonatal Surgery Unit: a Retrospective Study" raises the important issue of the dynamic spread of microorganisms resistant to some of the most important antibiotics used in the treatment of patients. The manuscript is well written and correctly presents the results obtained.
To further increase the quality of the manuscript, I suggest including a few modifications:
1. Lines 49-51: Problem with sentence order, and thus with understanding its meaning. I suggest a modification - especially since this is actually “the opening sentence” of the article.
2. Lines 56 + 58: Monobactams and carbapenems belong to beta-lactams, while both of these sentences imply that they are from other groups. Please modify (e.g. by using the word "including").
3. Line 73: detected -> isolated
4. Line 74: “… who backed from a travel in India and resulted colonized and infected by K. pneumoniae and E. coli” -> who backed from a travel in India infected by K. pneumoniae and E. coli
5. Lines 97-98: “and when necessary, decolonization is the current prevention strategy implemented to monitor carriage and infection” -> and when necessary, decolonization, are the current prevention strategies implemented to monitor carriage and infection
6. Lines 139-151: I wonder if the abstract and methodology should not indicate a longer period of observation. At the moment, at first glance, it appears that the research lasted about 3 months. Which is actually not true, because as described in lines 145-151 observations were carried out many months later.
7. I think it is worth increasing the size of the letters in the Figure 1, because after printing the article, you can hardly see anything.
8. Names of bacteria in the Figure 2 should be written using italics. Please modify.
9. Lines 236-237: “patient colonized with -NDM-producing E. coli” -> patient colonized with NDM-producing E. coli (please delete “-“ and modify the name of bacterium using italics)
10. Line 238: revealed -> turned out to be
11. Line 272: “pressure [35] therefore” -> pressure [35], therefore
Author Response
Response to Reviewer 2
- Lines 49-51: Problem with sentence order, and thus with understanding its meaning. I suggest a modification - especially since this is actually “the opening sentence” of the article.
We thanks the reviewer for this note. The sentence has been modified in order to improve its meaning. See lines 54-56.
- Lines 56 + 58: Monobactams and carbapenems belong to beta-lactams, while both of these sentences imply that they are from other groups. Please modify (e.g. by using the word "including").
Thanks for this note. Both sentences have been modified as suggested. See lines 59-61
- Line 73: detected -> isolated
This change has been made as indicated. See line 76
- Line 74: “… who backed from a travel in India and resulted colonized and infected by K. pneumoniae and E. coli” -> who backed from a travel in India infected by K. pneumoniae and E. coli
This sentence has been modified as suggested. See lines 76-78
- Lines 97-98: “and when necessary, decolonization is the current prevention strategy implemented to monitor carriage and infection” -> and when necessary, decolonization, are the current prevention strategies implemented to monitor carriage and infection.
This change has been made. See lines 106-109
- Lines 139-151: I wonder if the abstract and methodology should not indicate a longer period of observation. At the moment, at first glance, it appears that the research lasted about 3 months. Which is actually not true, because as described in lines 145-151 observations were carried out many months later.
Thanks to the reviewer for this note. We clarified in both abstract (see lines 31-32) and study design (see lines 345-347) sections that collection of samples was carried out between November 2020 and January 2021 followed by a longer period of observations due to the surveillance programme routinely applied within the Hospital. Thanks to this surveillance, we assessed that after a sporadic circulation resumed in April 2021, no cases were registered in 2022 until now.
- I think it is worth increasing the size of the letters in the Figure 1, because after printing the article, you can hardly see anything.
We thank the reviewer for this observation. In the revised version of the figure 1 the size of letters has been increased.
- Names of bacteria in the Figure 2 should be written using italics. Please modify.
Thanks to the reviewer for this observation. The bacteria names are in italics in the revised version of figure 2.
- Lines 236-237: “patient colonized with -NDM-producing E. coli” -> patient colonized with NDM-producing E. coli (please delete “-“ and modify the name of bacterium using italics)
Hyphen has been removed and E. coli is in italics. See lines 264-265
- Line 238: revealed -> turned out to be
This sentence has been modified as suggested. See line 266
- Line 272: “pressure [35] therefore” -> pressure [35], therefore
Comma has been added as suggested. See line 305
Reviewer 3 Report
The article, "Clonal Spread of Hospital-Acquired Colonization Cases by NDM-1-producing Klebsiella pneumoniae and Escherichia coli in an Italian Neonatal Surgery Unit: a Retrospective Study," reports the acquisition of NDM-1 carbapenemase-producing blaNDM-1 genes in Klebsiella pneumoniae and Escherichia coli in a neonatal surgical unit (NSU) at Bambino Gesù Children’s Hospital in Rome, Italy. The study involved stool sampling from neonates over a period of 64 days, followed by antibiotic susceptibility analysis, molecular characterization based on PCR-based replicon 32 typing (PBRT), and multilocus-sequence typing (MLST).
Major Comments:
-
The rationale for the chosen sampling period is not explained, and it may be significant to know if the timing has any relation to the seasonal occurrence or spread of infections or events.
-
The sample (stool) sampling and storage conditions must be detailed to understand the overall procedure implemented in this study.
-
The reason for selecting the specific gene, blaNDM-1, for this study must be provided, and references should be cited to support the selection. Additionally, the reason for only selecting K. pneumoniae and E. coli strains must be explained.
-
The authors should clarify whether the transmission of the blaNDM-1 gene is associated with the geographical location of the neonates or if it is related to hospital-acquired colonization alone. The criteria used to elucidate their findings must be explained, since phage transduction (horizontal gene transfers) could also provide such resistance genes.
-
The authors should consider monitoring food habits, such as breastfeeding, to determine whether there is any association between breastfeeding and such transmission.
Minor comments:
-
The introduction must be improved to clearly explain all the issues discussed above to reach a fair conclusion.
-
The authors should reconsider the title since it can be "Clonal Spread of Hospitalized Cases with NDM-1-producing Klebsiella pneumoniae and Escherichia coli" rather than "Hospital-Acquired Colonization Cases." This would require a lot of explanation in the results and discussion section to further support such findings
Author Response
Response to Reviewer 3
Major Comments:
- The rationale for the chosen sampling period is not explained, and it may be significant to know if the timing has any relation to the seasonal occurrence or spread of infections or events.
Thanks for this comment. Observations described in the present study come from the routine surveillance activity in place since 2012 within Bambino Gesù Children’s Hospital to monitor the prevalence of colonization/infection with multidrug-resistant Gram-negative microorganisms. The result of this activity is the frequent reporting of MDROs by microbiology laboratory. In this study, we wanted to highlight the sudden clonal spread observed in a specific ward (Neonatal Surgery Unit, NSU) in a very short period of time. Furthermore, no reoccurrence of other colonization cases was observed until now. For all of these reasons, no relation between the sampling period and seasonal occurrence of the described events can be reported.
- The sample (stool) sampling and storage conditions must be detailed to understand the overall procedure implemented in this study.
Stool sampling and storage conditions were not reported because samples were promptly processed upon their arrival at the microbiology laboratory. All isolated strains come from routinary procedure of the Hospital, as described in the “Hospital setting and preventive measures” paragraph. See section starting from the line 360.
- The reason for selecting the specific gene, blaNDM-1, for this study must be provided, and references should be cited to support the selection. Additionally, the reason for only selecting K. pneumoniae and E. coli strains must be explained.
We thank the reviewer for this observation. However, we didn’t select the specific blaNDM-1 gene, nor the K. pneumoniae and E. coli species for this study. Our findings derived from the systematic collection of data through observation emerging from the routine surveillance activity carried out by the Hospital. This monitoring detected the circulation of 20 CPEs strains with similar molecular profiles belonging to only K. pneumoniae and E. coli species, all positive for blaNDM-1 gene. All of these findings supported the hypothesis of a clonal spread of these species within the ward. We added this clarification in line 124 before the table 1 that shows epidemiological characteristics of patients, including isolated strains.
- The authors should clarify whether the transmission of the blaNDM-1 gene is associated with the geographical location of the neonates or if it is related to hospital-acquired colonization alone. The criteria used to elucidate their findings must be explained, since phage transduction (horizontal gene transfers) could also provide such resistance genes.
The study was focused on the investigation of molecular features of isolated strains in order to assess their clonal spread in relation to admission/hospitalization of patients in the same ward for a short period of time. Hence, the potential transmission of blaNDM-1 gene was related to hospital-acquired colonization. Furthermore, we have not studied other relations such as geographical location of neonates because the limited size of strains cannot be useful to obtain significant information about.
Findings discusses in our study concerning gene transmission derived from the observation that blaNDM-1 is frequently associated with plasmid profiles detected in our collection. WGS would be useful to assess this issue, and we discussed the limits of our study in lines 316-320.
- The authors should consider monitoring food habits, such as breastfeeding, to determine whether there is any association between breastfeeding and such transmission.
We thank the reviewers for this observation. As described in “Discussion” section, alternative sources of contamination and transmission could not be investigated (See lines 280-282). However, as suggested also by another reviewer we further clarified these limitations. See lines 282-285.
Minor comments:
- The introduction must be improved to clearly explain all the issues discussed above to reach a fair conclusion.
Thank you for this suggestion. Molecular mechanisms of carbapenem-resistance and characteristics of CPE strains were described in the first part of introduction. So, we better explained the occurrence of infections in children caused by CPE strains and provided more information about the relation between risk factors associated with the dissemination of resistant strains and the need to perform active surveillance in hospital. See lines 89-104
- The authors should reconsider the title since it can be "Clonal Spread of Hospitalized Cases with NDM-1-producing Klebsiella pneumoniae and Escherichia coli" rather than "Hospital-Acquired Colonization Cases." This would require a lot of explanation in the results and discussion section to further support such findings.
Considering also the suggestion of another reviewer we modified the title as the following: “Clonal Spread of Hospital-Acquired NDM-1-producing Klebsiella pneumoniae and Escherichia coli in an Italian Neonatal Surgery Unit: a Retrospective Study”.
Reviewer 4 Report
Authors should re-write this title for more clarity to readers. I would suggest simply writing this as, “Clonal Spread of Hospital-Acquired NDM-1-producing Klebsiella pneumoniae and Escherichia coli in an Italian Neonatal Surgery Unit: a Retrospective Study”
Introduction:
Line #72: Particular concern was for the New Delhi metallo-ß-lactamase (NDM), firstly detected in 2008 from a Sweden patient who backed from a travel in India and resulted colonized and infected by K. pneumoniae and E. coli. This sentence is not clear, consider re-writing it.
Line #118: All patients resulted positive to CPEs colonization during their hospitalization, since stool culture was negative for MDROs colonization at the time of their admission, but it resulted positive after 48h of hospital staying (starting from a minimum time of 6 days up to 212 days from hospitalization). Please re-write this sentence, All means how many patients here. Furthermore, Clearly distinguish between hospitalization and admission samples especially in this particular sentence.
Line #126: The negativization of 126 surveillance stool culture occur on the 21st of December. Consider re-writing this sentence.
Line # 190: This is written as the majority of iso- 190 lates (n = 10/20) were characterized by the multi-replicon status carrying three or more 191 different Inc groups. This is not majority instead it is half of the isolates.
Line #208: Five out of the seven PBRT profiles were characterized by a multireplicon status because composed by three or more Inc groups. Consider re-writing this sentence for clarity.
Line #222: In the last decades, several screening programmes were implemented in clinical settings worldwide, and many authors reported cases of colonization highlighting the serious implications for those patients considered at risk because fragile or immunocompromised [25–27]. Consider re-writing this sentence.
Line #234: of December 2020) was the time indicative of an ongoing clonal diffusion, which spreads in the following days. Consider re-writing this sentence.
Line #296 4.1 Ethics statementsOnly bacterial isolates recovered from routine screening and diagnostic laboratory tests were assessed in this study without direct use of clinical specimens. Considering the retrospective nature of the analysis, the current study did not require the approval of the local ethics committee according to current legislation, but a notification was sent. Data were retrospectively analyzed in line with personal data protection policies and patient consent was not required.
In light of the above statements, this study needs to obtain an ethical clearance. That clearly grants waiver of consent to use the study isolates and also access the presented patient clinical data as stated in Line #310 (We collected demographic and clinical information from electronic medical records about patients which resulted positive for CPE intestinal colonization, including: sex, age, geographical origin, the date of stool sample collection, and the preventive misures adopted to avoid the occurrence of infection)
Line #313: After isolation and identification, all CPE strains were characterized by antimicrobial susceptibility testing, detection of carbapenemase resistance determinants and PCR-based replicon typing (PBRT). Finally, to evaluate their clonality, all strains of K. pneumoniae and E. coli were analyzed by multi- locus-sequence typing (MLST). Authors should include the name of the laboratory where these assays were performed. Same in line #348, 375, 395, 411
Authors need to discuss the limitation of their methodology in confirming the clonal diffusion of NDM since they did not perform Whole-genome sequencing of the isolates.
Author Response
Response to Reviewer 4
- Authors should re-write this title for more clarity to readers. I would suggest simply writing this as, “Clonal Spread of Hospital-Acquired NDM-1-producing Klebsiella pneumoniae and Escherichia coli in an Italian Neonatal Surgery Unit: a Retrospective Study”
The title of the manuscript has been modified as suggested.
- Line #72: Particular concern was for the New Delhi metallo-ß-lactamase (NDM), firstly detected in 2008 from a Sweden patient who backed from a travel in India and resulted colonized and infected by K. pneumoniae and E. coli. This sentence is not clear, consider re-writing it.
As suggested also by another reviewer, we modified the sentence as the following: “Particular concern was for the New Delhi metallo-ß-lactamase (NDM), firstly isolated in 2008 from a Sweden patient who backed from a travel in India infected by K. pneumoniae and E. coli.” See lines 75-79
- Line #118: All patients resulted positive to CPEs colonization during their hospitalization, since stool culture was negative for MDROs colonization at the time of their admission, but it resulted positive after 48h of hospital staying (starting from a minimum time of 6 days up to 212 days from hospitalization). Please re-write this sentence, All means how many patients here. Furthermore, clearly distinguish between hospitalization and admission samples especially in this particular sentence.
The sentence has been clarified (see lines 139-143). Moreover, considering the suggestion of another reviewer, information about patients number, sex, age, hospital staying time (admission to discharge), date of isolate identified, were presented clearly in Table 1 of revised version.
- Line #126: The negativization of surveillance stool culture occur on the 21st of December. Consider re-writing this sentence.
This sentence has been rewrote as the following: “The negative evaluation of the monitoring stool culture takes place on December 21.” See lines 148-149
- Line # 190: This is written as the majority of iso- 190 lates (n = 10/20) were characterized by the multi-replicon status carrying three or more 191 different Inc groups. This is not majority instead it is half of the isolates.
Thanks for this observation. We rewrote the sentence specifying that it is half of the strains and not of most. See line 214
- Line #208: Five out of the seven PBRT profiles were characterized by a multireplicon status because composed by three or more Inc groups. Consider re-writing this sentence for clarity.
We agree with the reviewer. This sentence has been rewrote as the following: “Five PBRT profiles were composed by three or more Inc groups. See lines 232-233
- Line #222: In the last decades, several screening programmes were implemented in clinical settings worldwide, and many authors reported cases of colonization highlighting the serious implications for those patients considered at risk because fragile or immunocompromised [25–27]. Consider re-writing this sentence.
We thank the reviewer for the note. The sentence has been modified as following: “During recent decades, several screening programs were implemented in clinical settings world-wide. Many cases of colonization were described highlighting the risk for fragile or immunocompromised patients.” See lines 249-251
- Line #234: of December 2020) was the time indicative of an ongoing clonal diffusion, which spreads in the following days. Consider re-writing this sentence.
Thanks to the reviewer for this note that allowed us to improve this sentence. It has been modified as the following: “In the present study, the detection of three patients positive for NDM-intestinal colonization on the 14th of December 2020 was the starting point of clonal spread.” See lines 259-260
- Line #296 4.1 Ethics statements. Only bacterial isolates recovered from routine screening and diagnostic laboratory tests were assessed in this study without direct use of clinical specimens. Considering the retrospective nature of the analysis, the current study did not require the approval of the local ethics committee according to current legislation, but a notification was sent. Data were retrospectively analyzed in line with personal data protection policies and patient consent was not required.
In light of the above statements, this study needs to obtain an ethical clearance. That clearly grants waiver of consent to use the study isolates and also access the presented patient clinical data as stated in Line #310 (We collected demographic and clinical information from electronic medical records about patients which resulted positive for CPE intestinal colonization, including: sex, age, geographical origin, the date of stool sample collection, and the preventive misures adopted to avoid the occurrence of infection)
We thank the reviewers for this observation. At the time of hospitalization, the parents or legal guardians of the patients consent to the use of personal data for diagnosis and treatment activities and also for future scientific research purposes connected to them. We added this aspect in Ethics statements of revised version (See lines 338-340). Basic demographic and clinical information collected in this study, did not infringe upon the rights or welfare of the patients. For these reasons, the approval of the local ethics committee was not necessary.
- Line #313: After isolation and identification, all CPE strains were characterized by antimicrobial susceptibility testing, detection of carbapenemase resistance determinants and PCR-based replicon typing (PBRT). Finally, to evaluate their clonality, all strains of K. pneumoniae and E. coli were analyzed by multi- locus-sequence typing (MLST). Authors should include the name of the laboratory where these assays were performed. Same in line #348, 375, 395, 411
Thank you for this suggestion. All assays were carried out by the staff of Bambino Gesù Children’s Hospital and Urbino University. We clarified this point in the section of study design adding the related laboratory for each assays mentioned. See lines 356-359.
- Authors need to discuss the limitation of their methodology in confirming the clonal diffusion of NDM since they did not perform Whole-genome sequencing of the isolates.
We thank the reviewer for this note. We clarify that a limitation of our investigation is the clonal spread of strains investigated by MLST that is less discriminating and informative than WGS. See lines 316-320.
Round 2
Reviewer 3 Report
The author's have responded to all the comments and now the article is much more clear and can be considered for publication.
Reviewer 4 Report
None